# Beyond Ion Homeostasis: Hypomagnesemia, Transient Receptor Potential Melastatin Channel 7, Mitochondrial Function, and Inflammation

**DOI:** 10.3390/nu15183920

**Published:** 2023-09-09

**Authors:** Man Liu, Samuel C. Dudley

**Affiliations:** Cardiovascular Division, Department of Medicine, The Lillehei Heart Institute, University of Minnesota at Twin Cities, Minneapolis, MN 55455, USA; sdudley@umn.edu

**Keywords:** cardiovascular diseases, oxidative stress, inflammation, therapeutics, mitochondria

## Abstract

As the second most abundant intracellular divalent cation, magnesium (Mg^2+^) is essential for cell functions, such as ATP production, protein/DNA synthesis, protein activity, and mitochondrial function. Mg^2+^ plays a critical role in heart rhythm, muscle contraction, and blood pressure. A significant decline in Mg^2+^ intake has been reported in developed countries because of the increased consumption of processed food and filtered/deionized water, which can lead to hypomagnesemia (HypoMg). HypoMg is commonly observed in cardiovascular diseases, such as heart failure, hypertension, arrhythmias, and diabetic cardiomyopathy, and HypoMg is a predictor for cardiovascular and all-cause mortality. On the other hand, Mg^2+^ supplementation has shown significant therapeutic effects in cardiovascular diseases. Some of the effects of HypoMg have been ascribed to changes in Mg^2+^ participation in enzyme activity, ATP stabilization, enzyme kinetics, and alterations in Ca^2+^, Na^+^, and other cations. In this manuscript, we discuss new insights into the pathogenic mechanisms of HypoMg that surpass previously described effects. HypoMg causes mitochondrial dysfunction, oxidative stress, and inflammation. Many of these effects can be attributed to the HypoMg-induced upregulation of a Mg^2+^ transporter transient receptor potential melastatin 7 channel (TRMP7) that is also a kinase. An increase in kinase signaling mediated by HypoMg-induced TRPM7 transcriptional upregulation, independently of any change in Mg^2+^ transport function, likely seems responsible for many of the effects of HypoMg. Therefore, Mg^2+^ supplementation and TRPM7 kinase inhibition may work to treat the sequelae of HypoMg by preventing increased TRPM7 kinase activity rather than just altering ion homeostasis. Since many diseases are characterized by oxidative stress or inflammation, Mg^2+^ supplementation and TRPM7 kinase inhibition may have wider implications for other diseases by acting to reduce oxidative stress and inflammation.

## 1. Introduction

As the second most abundant intracellular divalent cation, magnesium (Mg^2+^) is essential for cell functions, such as ATP production, protein and DNA synthesis, protein activity, and mitochondrial function. Mg^2+^ is a co-factor of hundreds of enzymes, including ATPases, endonucleases, exonucleases, polymerases, and phosphatases [1]. Mg^2+^ is also a calcium (Ca^2+^) antagonist, playing an important role in regulating Ca^2+^ homeostasis and function [2,3]. Traditionally, the effects of changes in Mg^2+^ levels have been viewed as general alterations in these Mg^2+^-dependent processes or through effects on the homeostasis of other ions such as Ca^2+^ or sodium ions (Na^+^). 

In this review, we discuss new data that suggest Mg^2+^ to be in a transcriptional feedback loop with a Mg^2+^ transporter, transient receptor potential melastatin 7 channel (TRPM7), which also has a kinase domain. These new data raise the possibility that much of the antioxidant and anti-inflammatory effects of Mg^2+^ supplementation are mediated by changes in this channel/kinase level and its subsequent effects on kinase signal activity independent of any effects on Mg^2+^ ion distribution.

## 2. Hypomagnesemia and Its Association with Disease

New data on the role of TRPM7 have come to light because of experiments that study hypomagnesemia (HypoMg, serum Mg^2+^ < 0.8 mM) [4]. It seems increasingly clear that HypoMg is more than just a generalized dysfunction of Mg^2+^-dependent processes. HypoMg is a disease condition that is characterized by, among other things, increased mitochondrial dysfunction and inflammation that can lead to a type of heart failure and seizures [5,6]. 

HypoMg is more common than is generally known, and its incidence is increasing for several reasons. A significant decline in Mg^2+^ intake has been reported in developed countries because of the increased consumption of processed food and filtered/deionized water [7,8]. Studies show that ~50% of the US population, especially elderly people, consume less than the estimated average requirement of Mg^2+^ [9,10,11], and ~23% of US adults have hypomagnesemia (HypoMg, serum Mg < 0.8 mM) [4]. Moreover, chronic kidney, gastrointestinal diseases, and medications (e.g., diuretics, cytotoxic drugs, digoxin, amino-glycosides, and steroids) can further increase Mg^2+^ excretion and decrease Mg^2+^ absorption, causing HypoMg. A larger driver of the increase in HypoMg is likely an increase in diabetes. HypoMg and diabetes are epidemiologically associated [12,13,14,15], and with the worldwide increase in diabetes, we are likely to see a concomitant increase in HypoMg.

HypoMg is commonly observed in a variety of illnesses, including cardiovascular diseases [16,17,18,19,20], epilepsy, Alzheimer’s disease, Parkinson’s disease, osteoporosis, and diabetes (see reviews [21,22,23]). The cause-and-effect relationship between these disease states and HypoMg is an area of active investigation, and it is possible that many of these disease states actively contribute to HypoMg rather than being exacerbated by the condition.

HypoMg has been reported in heart failure [24,25], arrhythmia [26,27,28], myocardial infarction [27,28], atherosclerosis [29], diabetic cardiomyopathy [14,15], hypertension [20,30,31], and stroke [32], and Mg^2+^ supplementation has been used to treat some of these conditions. For example, patients with congestive heart failure often show signs of HypoMg [24,25], while Mg^2+^ supplementation has protective effects against heart failure [24]. HypoMg is reported in nearly 40% of patients with ventricular arrhythmias at hospital admission, and Mg^2+^ supplementation significantly decreases the number of episodes of non-sustained ventricular tachycardia [26]. In patients with acute myocardial ischemia, low serum Mg^2+^ has been observed, while Mg^2+^ supplementation improves heart function and mortality [27,28,33]. HypoMg and a low Mg^2+^ diet are associated with a higher risk of coronary artery disease, atherosclerosis, and hypertension [33,34,35], while Mg^2+^ supplementation improves these conditions [35,36,37].

## 3. Canonical Roles of Mg^2+^ in Cardiovascular Disease

The effects of HypoMg have generally been ascribed to alterations in Mg^2+^ or other cations in various subcellular compartments, and the effects of Mg^2+^ supplementation are thought to be mediated by reversing these generalized changes. 

Mg^2+^ homeostasis is essential for cardiovascular function, including myocardial metabolism, myocardial contraction and relaxation, cardiac output, vascular tone, and peripheral vascular resistance [5,13,21,22,23]. In cardiomyocytes, the total intracellular Mg^2+^ concentration ([Mg^2+^]_i_) is ~17–20 mmol/L. Much of this Mg^2+^ is not free. About 50% of intracellular Mg^2+^ binds to ATP, phosphonucleotides, and phosphometabolites, while 25% binds to ribosomes [38]. Resting, free Mg^2+^ is in the range of 0.8–1.0 mmol/L both in the cytoplasm [39] and in the matrix of cardiomyocyte mitochondria [40]. Mg^2+^ regulates multiple cardiac ion channels, including the Na^+^ channel, multiple K^+^ channels, L- and T-type Ca^2+^ channels, the Na^+^-Ca^2+^ exchanger, Na^+^-K^+^-ATPase, and sarcoplasmic reticulum Ca^2+^ release, accounting for some of Mg^2+^’s effects on other cations (see review [22,41]). Mg^2+^ is also a critical cofactor for ATP production in cardiac mitochondria.

The disturbance of Mg^2+^ homeostasis can affect intracellular ion concentrations and balance, alter myocyte membrane potential, and impair heart contractility and heart rhythm. Mg^2+^ homeostasis is also important for vascular function and blood pressure. Decreased Mg^2+^ leads to vasoconstriction, while increased Mg^2+^ causes vasodilation and suppresses agonist-induced vasoconstriction [42,43]. Mechanistic studies show that in vascular smooth muscle cells, Mg^2+^ regulates excitation–contraction coupling and the myosin–actin crossbridge by modulating Ca^2+^ homeostasis [17,41,44]. 

Decreased [Mg^2+^]_i_ is also associated with pulmonary hypertension, and high Mg^2+^ prevents the upregulated expression of NFATc3 and its nuclear translocation, which is a major Ca^2+^-dependent signaling pathway that regulates the proliferation and migration of pulmonary arterial smooth muscle cells in the development of pulmonary hypertension and vascular remodeling [35]. This is an important observation because it links Mg^2+^ and gene transcription. HypoMg also affects vascular endothelial cells’ survival, proliferation, and motility and induces higher vascular–endothelial barrier permeability [45]. Hormones and vasoactive agents such as angiotensin II, insulin, aldosterone, and vasopressin that are often altered in cardiovascular diseases also affect free [Mg^2+^]_i_ in cardiomyocytes and vascular smooth muscle cells [44].

## 4. Mg^2+^ Transporters Control Mg^2+^ and Link Mg^2+^ to Other Cations

Mg^2+^ transporters are critical for maintaining Mg^2+^ homeostasis in cells and intracellular organelles. They also link Mg^2+^ homeostasis with other cations. Cardiac Mg^2+^ homeostasis is regulated and maintained by a series of sarcolemmal and organelle transporters, such as the TRPM7, solute carrier family 41 A1 (SLC41A1), Mg^2+^ transporter 1 (MagT1), and cyclin and CBS domain divalent metal cation transport mediator 2 (CNNM2) on the sarcolemmal membrane. In addition, there is the mitochondrial RNA splicing 2 protein (MRS2) and solute carrier family 41 A3 (SLC41A3) on mitochondrial membranes, as well as some antiporters, cotransporters, and exchangers (see review [22,46,47]). It is these antiporters, cotransporters, and exchangers that link Mg^2+^ homeostasis to that of other cations.

Certainly, Mg^2+^ transporters need more study, but some key facts are known. The SLC41A family has three members identified as Mg^2+^ transporters/carriers: SLC41A1, SLC41A2, and SLC41A3 (see a recent review [48]). SLC41A1 and SLC41A2 are expressed mainly on the plasma membrane [49,50], while SLC41A3 is mainly expressed on mitochondrial membranes and conducts mitochondrial Mg^2+^ efflux [51]. SCL41A1 conducts Mg^2+^ influx [52] or Na^+^-dependent Mg^2+^ efflux [49]. MRS2 regulates mitochondrial Mg^2+^ influx [53]. MagT1 shows strong specificity for Mg^2+^ with a voltage dependence [54] and localizes in the plasma and organelle membranes of endoplasmic reticulum and Golgi [55]. MagT1 regulates vascular endothelial survival, proliferation, and motility [45]. CNNM2 conducts Mg^2+^ uptake in a voltage-dependent manner [56] and can also regulate TRPM7-mediated Mg^2+^ transport [57], suggesting that these transporters can act in concert and feedback to each other.

These Mg^2+^ channels and transporters are not always specific to Mg^2+^ and can be permeable to multiple divalent cations with a selectivity that varies among the specific transporters. For example, TRPM7 also mediates Ca^2+^ transport, while the three SLC41A family isoforms do not transport Ca^2+^.

## 5. HypoMg Is More than Just Alterations in Ion Homeostasis

It is becoming clear that the changes with HypoMg are more specific than just the alterations in divalent cations. Recently, our group reported that Mg^2+^ modulates myocardial relaxation through regulating mitochondrial oxidative stress; HypoMg, in the presence or absence of diabetes mellitus, can cause mitochondrial reactive oxygen species overproduction (oxidative stress) and lead to the inability of the heart to properly relax and a type of heart failure known as diastolic heart failure, while Mg^2+^ supplementation or repletion can reverse these changes [5,13]. In these studies, the effects of HypoMg went beyond just having less cell energy, a loss of competition with Ca^2+^, and the disturbance of a divalent cation balance across the plasma membrane. HypoMg resulted in specific defects in the mitochondrial electron transport chain, increased mitochondrial oxidative stress, and mitochondrial membrane potential depolarization [5,13,41,58,59,60].

HypoMg is related to increased inflammation [61], either caused by or resulting in mitochondrial reactive oxygen species overproduction [5,62], in what likely constitutes a vicious cycle. Our recent studies show that HypoMg induces inflammation [6], which leads to mitochondrial oxidative stress and cardiac diastolic dysfunction or the failure of the heart to properly relax. This lack of relaxation can progress to a unique form of heart failure: heart failure with preserved ejection fraction (HFpEF) or diastolic heart failure, which is when the heart cannot pump enough blood because it does not fill sufficiently during the relaxation phase [63]. This type of heart failure has similar mortality and symptoms to that of heart failure, where the pump function is impaired, known as systolic heart failure. The difference in pathogenesis between these two types of heart failure is emphasized by the fact that HFpEF does not respond to therapies that are effective for systolic heart failure, and there are no known therapies for HFpEF that affect its poor mortality rate.

In an animal model of HFpEF, cardiac inflammation is manifested as increased interleukin-1β (IL-1β) levels and an altered cardiac macrophage phenotype with increased pro-inflammatory macrophages and decreased anti-inflammatory, pro-resolving macrophages [63]. IL-1β elevation triggers cardiomyocyte mitochondrial oxidative stress, causes the oxidation of cardiac myosin-binding protein C, and leads to cardiac diastolic dysfunction [63]. An IL-1 receptor antagonist, a mitochondrial-targeted antioxidant, or Mg^2+^ supplementation can improve mitochondrial oxidative stress and cardiac diastolic dysfunction caused either by HypoMg or diabetes [5,13,63], suggesting that HypoMg mediates this form of heart failure associated with diabetes.

## 6. TRPM7 May Mediate the Oxidative Stress and Inflammation of HypoMg

Among the Mg^2+^ transporters, TRPM7 has been extensively characterized. It regulates both whole-body Mg^2+^ homeostasis [64,65] and cellular Mg^2+^ homeostasis in many types of cells, including cardiomyocytes, vascular smooth muscle cells, and B lymphoma cells [66,67,68], but not T cells [69]. TRPM7 is highly permeable for Mg^2+^, Ca^2+^, Zn^2+^, Mn^2+^, and a few other divalent cations. TRPM7 both regulates and is regulated by Mg^2+^. TRPM7 currents are inhibited by the physiological concentration of Mg^2+^ and Mg-ATP. 

In addition to its transport function, TRPM7 is a unique ion channel in that it has an α kinase domain that regulates protein transcription [70,71] and phosphorylation [72,73,74,75,76], as well as a sensor for Mg^2+^’s status [66]. TRPM7 is essential for early cardiogenesis [77], cardiac repolarization [77], cardiac automaticity and rhythmicity [78], vascular endothelial survival, proliferation and motility [45], and vascular smooth muscle cell growth [79].

The TRPM7 protein amount is counter-regulated to the Mg^2+^ concentration. That is to say, HypoMg increases TRPM7 transcription and translation. Increased TRPM7 is associated with increased kinase signaling. Under HypoMg, the upregulation of TRPM7 can be observed, including elevated TRPM7 mRNA translation [80], protein expression [6,45], and TRPM7-conduced Mg^2+^ currents [81,82,83]. A recent animal study on pulmonary hypertension shows an association between decreased [Mg^2+^]_i_ in pulmonary arterial smooth muscle cells and the upregulation of TRPM7 expression, as well as the upregulation of SLC41A1, SCL41A2, CNNM2, MRS2, and MagT1 [35]. Furthermore, the upregulated TRPM7 function has been observed in patients with ischemia-reperfusion, arrhythmias, and hypertension [84,85,86].

Many of the effects of HypoMg can be explained by elevated TRPM7 and the activation of the TRPM7 kinase. For example, the inflammation and oxidative stress that are associated with HypoMg are correlated with TRPM7 expression and function. Our recent work shows that TRPM7 kinase activation via HypoMg contributes to the inflammation activation and oxidative stress observed in HypoMg-induced seizure activity, all of which can be significantly improved in transgenic mice with a K1646R mutation in the kinase domain that results in no kinase function while leaving transport function intact [6].

How TRPM7 kinase alters mitochondrial oxidative stress alongside inflammation is an area of active investigation. TRPM7-mediated Ca^2+^ signaling enhances macrophage activation in response to provocation. When selectively depleted in bone marrow macrophages, inflammatory marker IL-1β secretion was significantly reduced, and mice were resistant to peritonitis induced by a classic inflammatory mediator derived from bacteria [87]. Silencing TRPM7 improves inflammation and apoptosis in renal ischemic reperfusion injury, while the activation of TRPM7 exacerbates hypoxia/reoxygenation-induced renal injury [88].

What controls TRPM7 levels is also an area that needs more investigation. The inflammatory mediator IL-18 upregulates TRPM7 expression and currents in vascular smooth muscle cells [89], and angiotensin II and aldosterone, as mediators of hypertension and promoters of atherosclerosis, have been shown to activate TRPM7 to induce Mg^2+^ influx and oxidative stress production in vascular smooth muscle cells [75,79,90].

These data suggest a positive feedback loop between TRPM7 and inflammation. There is also a positive feedback loop between oxidative stress levels and TRPM7’s expression and function. Increased oxidative stress levels elevate TRPM7 expression and currents [91,92], and TRPM7 overexpression induces intracellular oxidative stress overproduction [6,93,94]. A similar positive feedback loop exists between oxidative stress and HypoMg in cardiomyocytes [5,95]. TRPM7 elevation with HypoMg could underlie this correlation between oxidative stress and HypoMg.

Other TRPM channels likely play synergistic roles in Mg^2+^ homeostasis and other deleterious effects of HypoMg. TRPM6, like TRPM7 with both the channel and kinase function, is predominantly expressed in the kidney, cecum, and colon. It plays an important role in Mg^2+^ absorption in the gastrointestinal tract and Mg^2+^ excretion in the kidney [96,97,98]. TRPM6 is also upregulated under HypoMg or oxidative stress [98,99,100]. This channel’s role in cardiovascular diseases is unclear, but it may amplify the effects of HypoMg or even be a primary cause of HypoMg. For example, TRPM6 upregulation may be important when understanding the association between diabetes and HypoMg [65,96,101,102]. Moreover, TRPM6 phosphorylates TRPM7 [103] and can, therefore, modulate TRPM7 activity in cardiovascular diseases. However, the full extent of the interplay between these two TRPM channels is unknown.

## 7. Treatment of HypoMg-Related and Other Inflammatory Disorders

The inverse relationship between TRPM7 and Mg^2+^ levels supports the use of Mg^2+^ supplementation to counteract the effect of HypoMg-related pathology. The role of TRPM7 in this pathology suggests that kinase inhibition may be another therapeutic approach. In this regard, dietary Mg^2+^ intake is inversely correlated to cardiovascular events, such as increased risk of heart failure, hospitalization [104], coronary heart disease [34], hypertension [20,30,31], stroke [12,32], and diabetic cardiomyopathy [12,13]. An obvious disease candidate in need of further investigation related to these proposed mechanisms is pre-eclampsia, where Mg^2+^ is well-established clinically as a therapy. Nevertheless, it is unclear whether Mg^2+^ supplementation in pre-eclampsia demonstrates some or all of its effects through the modulation of TRPM7, oxidative stress, and inflammation.

Aside from genetic approaches, the modulation of TRPM7 is a possibility in humans with small molecule therapies. For TRPM7, there are inhibitors for channel activity (e.g., NS8593, waixenicin A, and FTY720) and for the kinase function, such as TG100-115 (see review [105,106]). TG100-115 has been shown to restrict the infarct size after myocardial ischemia-reperfusion injury [107].

## 8. Unknowns

The discussion above suggests that there is more to the effect of Mg^2+^ supplementation than just generalized effects on enzyme kinetics and ion homeostasis, though there are many unknowns. Some of these are: How many diseases characterized by inflammation and oxidative stress respond to Mg^2+^ therapy? Do these diseases need to be associated with HypoMg or TRPM7 elevation to respond? Are specific cell types, such as white cells, mediating the effects of HypoMg, or is the response generalized to most cells? To what extent does TRPM7 kinase activity explain the many effects of HypoMg, and to what extent are cation changes important? What are the downstream effectors activated by TRPM7 kinase, and how does the kinase alter mitochondrial function? How are mitochondrial dysfunction and inflammation linked? How do reductions in Mg^2+^ result in TRPM7 transcriptional upregulation, and could this signaling cascade be interrupted? Can oral Mg^2+^ supplementation be effective for treating Mg^2+^-responsive diseases in humans, and can sufficient doses be achieved without inducing gastrointestinal side effects? Could TRPM7 kinase inhibition be a solution to the issues related to oral Mg^2+^ supplementation?

## 9. Conclusions

Above, we have discussed the idea that the effects of HypoMg are more than just a general loss of Mg^2+^ and the related shifts in other cations and their dependent processes. We have proposed that many of the effects of HypoMg could be explained by a signaling cascade involving the TRPM7 kinase, which is upregulated by HypoMg. We have argued that Mg^2+^ supplementation or TRPM7 kinase inhibition can prevent HypoMg-induced mitochondrial oxidative stress and inflammation and that these treatments may be useful in such conditions as diabetes, where Mg^2+^ deficiency is common. We have indicated that Mg^2+^ may be a simple therapy for a common type of heart failure: HFpEF. Moreover, we have speculated that these therapies could be helpful in diseases characterized by oxidative stress and inflammation, but not known to be associated with HypoMg. We have pointed out just some of the unknowns when following this train of thought.

What is clear is that there is more to Mg^2+^ than is generally appreciated, and these subtleties could lead us to new understandings of diseases and potential new therapies.

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
