# Peer review of "Beyond Ion Homeostasis: Hypomagnesemia, Transient Receptor Potential Melastatin Channel 7, Mitochondrial Function, and Inflammation"

_nutrients, 2023, doi:10.3390/nu15183920_

Round 1
Reviewer 1 Report
This is an interesting opinion article. Certain concern should be addressed.
- The opinion article is quite condensed and enough difficult to be undestood by the potential redears
- A more simple language could be applied, providing some more explanation in certain biochemistry issues even if these are obvious for the authors.
- Some paragraphs are too long aand could be separated into two paragraphs.
- The authors could emphasize the gap of the literature in this topic
- At the end of the article, the authors could add more comprehensively their opinion for the future studies that shoud be perform to cover the literature gap of this topic.
- Minor English editing is required
-
-
Minor English editing is required.
Author Response
- The opinion article is quite condensed and enough difficult to be understood by the potential readers
Response: We appreciate the concern and have tried to make the article more readable..
- A more simple language could be applied, providing some more explanation in certain biochemistry issues even if these are obvious for the authors.
Response: As above, we appreciate the issue and have tried to make the article more approachable.
- Some paragraphs are too long and could be separated into two paragraphs.
Response: We have tried to focus each paragraph.
- The authors could emphasize the gap of the literature in this topic
Response: We agree. We have added a section on knowledge gaps entitled, “Unknowns.”.
- At the end of the article, the authors could add more comprehensively their opinion for the future studies that should be perform to cover the literature gap of this topic.
Response: With the “Unknowns” section and the summary, we have tried to address this concern.
- Minor English editing is required
Response: We have tried to proofread more carefully.
Reviewer 2 Report
The manuscript fits well within the intent of the special issue. It takes the reader across multiple levels. EPI, health/Clinical importance, and mechanistic.
As a general concern, the authors appreciate that the topic area is huge. The sprcific focus for this review should be made clear.
· The abstract is missing the main framing statement for the review to guide the reader. —suggest revising.
· The first paragraph mini-intro about the importance of magnesium can then re-iterate the framing for the review. —suggest revising
There currently isn’t a “methods” statement that indicates what level review this manuscript represents. — Suggest revising to include.
Currently, the review is a bit hard to follow easily. —Suggest adding subheadings to denote framing. The result will make it easier to construct the various key pieces, prevent redundancy and make for a clearer presentation.
Reviewer noticed that this recent review relevant to CVD is not mentioned https://pubmed.ncbi.nlm.nih.gov/33447739/
Minor
The second paragraph should begin with the EPI of intake. “A significant…. Moving the current first sentence after HypoMg is defined.
Ln 76 typo.. “again” — should be against
Author Response
- As a general concern, the authors appreciate that the topic area is huge. The specific focus for this review should be made clear.
- The abstract is missing the main framing statement for the review to guide the reader. —suggest revising.
Response: We agree. Now, we state, “In this manuscript, we discuss new insights into the pathogenic mechanisms of HypoMg that go beyond these previously described effects. HypoMg causes mitochondrial dysfunction, oxidative stress, and inflammation. Many of these effects can be attributed to HypoMg-induced upregulation of a Mg2+ transporter transient receptor potential melastatin 7 channel (TRMP7) that is also a kinase. An increase in the kinase signaling mediated by HypoMg-induced TRPM7 transcriptional upregulation, independently of any change in Mg2+ transport function, likely seems responsible for many of the effects of HypoMg. Therefore, Mg2+ supplementation and TRPM7 kinase inhibition may work to treat the sequelae of HypoMg by preventing increased TRPM7 kinase activity rather than just altering ion homoeostasis.”
- The first paragraph mini-intro about the importance of magnesium can then re-iterate the framing for the review. —suggest revising
Response: We agree. Now, we state, “In this review, we will discuss new data that suggest Mg2+ is in a transcriptional feedback loop with a Mg2+ transporter, transient receptor potential melastatin 7 channel (TRPM7), which also has a kinase domain. The new data raises the possibility that much of the antioxidant and anti-inflammatory effects of Mg2+ supplementation are mediated by changes in this channel/kinase level and its subsequent effects on kinase signaling activity independently of any effects on Mg2+ ion distribution.”
- There currently isn’t a “methods” statement that indicates what level review this manuscript represents. — Suggest revising to include.
Response: We have attempted to address this by being clearer about the purpose of our manuscript.
- Currently, the review is a bit hard to follow easily. —Suggest adding subheadings to denote framing. The result will make it easier to construct the various key pieces, prevent redundancy and make for a clearer presentation.
Response: We have added subheadings as highlighted. We thank you for the suggestion.
- Reviewer noticed that this recent review relevant to CVD is not mentioned https://pubmed.ncbi.nlm.nih.gov/33447739/
Response: We have added this valuable review as reference 20 at three places.
Minor
- The second paragraph should begin with the EPI of intake. “A significant…. Moving the current first sentence after HypoMg is defined.
Response: We have restricted this whole subhead and hope the logic flow is better now.
- Ln 76 typo.. “again” — should be against
Response: We have corrected this. We thank you for pointing it out.